# FISHNET++: ANALYZING THE CAPABILITIES OF MULTIMODAL LARGE LANGUAGE MODELS IN MARINE BIOLOGY

## ABSTRACT

Multimodal large language models (MLLMs) have demonstrated impressive cross-domain capabilities, yet their proficiency in specialized scientific fields like marine biology remains underexplored. In this work, we systematically evaluate state-of-the-art MLLMs and reveal significant limitations in their ability to perform fine-grained recognition of fish species, with the best open-source models achieving less than 10% accuracy. This task is critical for monitoring marine ecosystems under anthropogenic pressure. To address this gap and investigate whether these failures stem from a lack of domain knowledge, we introduce Fish-Net++, a large-scale, multimodal benchmark. FishNet++ significantly extends existing resources with 35,133 textual descriptions for multimodal learning, 706,426 key-point annotations for morphological studies, and 119,399 bounding boxes for detection. By providing this comprehensive suite of annotations, our work facilitates the development and evaluation of specialized vision-language models capable of advancing aquatic science.

## 1 INTRODUCTION

Healthy aquatic ecosystems and the services they provide are essential for human survival Selig et al. (2019); Basurto et al. (2025); Barbier (2017). The health of these ecosystems and the volume and quality of ecosystem services are closely tied to changes in their biodiversity Worm et al. (2006); Tett et al. (2013). At a time when aquatic ecosystems are under intense threat from human activities such as fisheries, climate change, coastal development, and pollution, conservation and management interventions are critical in preserving and restoring ecosystem health. Most conservation efforts begin with basic documentation, recognition, and monitoring of biodiversity; in aquatic ecosystems, these efforts are complicated by their often remote and relatively inaccessible nature. As a result, they become time and labor-intensive processes that require expert knowledge to undertake what might otherwise be considered relatively menial tasks. When extrapolated to the global scale, this first step presents a critical bottleneck in our ability to generate the information required to make informed decisions and to take the essential conservation and management actions required to preserve the health of aquatic ecosystems.

Recent advances in Multimodal Large Language Models (MLLMs) offer promising potential for automation across a variety of tasks, having demonstrated exceptional generalist skills in vision-language tasks Chen et al. (2022a); Alayrac et al. (2022); Singh et al. (2022); Liu et al. (2023b); Zhu et al. (2023); Chen et al. (2023). However, it is unclear if this proficiency translates to the fine-grained, expert-level knowledge required for marine species recognition to support conservation efforts.

To address this, we conduct a systematic analysis to answer a crucial question: Do state-of-the-art Multimodal Large Language Models (MLLMs) possess the specialized knowledge required for aiding marine ecology conservation efforts, or do their capabilities degrade when confronted with fine-grained, out-of-distribution data? We first probe the recognition level of leading MLLMs by evaluating their zero-shot species recognition performance, revealing that even the most capable models lack domain knowledge. Qwen2.5-VL achieves just 6.2% accuracy on frequent species and 0.2% on rare species.

This initial finding motivates a deeper diagnostic question: Does this failure stem from a core lack of domain-specific knowledge or from inadequate visual perception of fine-grained features in the marine domain? To disentangle these factors, we design three targeted tasks. 1) Domain Knowledge: We assess the models' domain knowledge by evaluating their ability to relate common names to scientific names and vice versa. 2) Visual Domain Knowledge: We evaluate the visual domain knowledge by testing whether the models can verify the presence/absence of a given species in an image. 3) Perception Capabilities: We test how well the models can (a) locate the species with a bounding box, and (b) pinpoint specific morphological structures through key-part localization.

To facilitate this investigation, we introduce FishNet++, a large-scale, multi-modal benchmark designed not only to diagnose these limitations but also to help improve recognition. FishNet++ comprises 99,556 images across 17,393 fish species, enriched with 706,426 key-point annotations, 119,399 bounding boxes, and detailed textual descriptions. We leverage this benchmark to first quantify the zero-shot performance of MLLMs and then show their lack of domain knowledge and how improvements can be achieved.

To summarize, our contributions are:

- We conduct the first large-scale analysis of MLLMs in the marine domain, revealing critical performance gaps in their zero-shot knowledge of marine species.
- We conduct a detailed diagnostic analysis, deconstructing this poor performance across three tasks to disentangle failures in semantic knowledge from visual perception.
- We introduce FishNet++, a comprehensive multi-modal benchmark with annotations for open-vocabulary recognition, detection, and keypoint localization, serving both as a diagnostic tool for evaluating MLLMs and as a resource for developing stronger marine domain-aware models.
- We demonstrate that the identified knowledge gap can be mitigated, showing that fine-tuning on FishNet++ substantially boosts MLLM performance.

## 2    RELATED WORK

**Open-Vocabulary Recognition.**    The task of open-vocabulary recognition has evolved from early works like Zhao et al. (2017), which introduced joint image–word embeddings for semantic segmentation, allowing models to go beyond fixed label sets. This line of research gained momentum with the advent of large-scale pretrained models such as BERT Devlin et al. (2019) for text and CLIP Radford et al. (2021), which aligned vision and language embeddings for zero-shot classification. CLIP's success led to extensions for open-vocabulary detection Gu et al. (2022), segmentation Li et al. (2022), and classification Dao et al. (2023); Zhu et al. (2024). While CLIP-like models Radford et al. (2021); Ilharco et al. (2021); Zhai et al. (2023) perform well in general settings, they remain suboptimal in fine-grained, open-world recognition, likely due to limited taxonomic understanding and dataset bias. This is discussed further in Section 4.2.

**Dense Recognition Tasks.**    Classical dense recognition methods rely on bounding-box or pixel-wise prediction. One-stage detectors like YOLO Redmon et al. (2016) unify localization and classification for real-time inference (up to 155 fps). Two-stage detectors such as Faster R-CNN Ren et al. (2016) generate region proposals before classification, and Mask R-CNN He et al. (2018) extends this by adding a segmentation branch. Transformer-based DETR Carion et al. (2020) reframes detection as set prediction using an encoder–decoder transformer, removing the need for non-maximum suppression and anchors. For segmentation, models like FCN Long et al. (2015), DeepLab Chen et al. (2017), MaskFormer Cheng et al. (2021), and SAM Kirillov et al. (2023b) demonstrate strong generalization. In fish imagery, these architectures (e.g., YOLO, Mask R-CNN) are widely applied with domain-specific tuning. Given its efficiency, we adopt YOLO-based Redmon et al. (2016) models for our dense-prediction tasks.

**Species Recognition.**    Fine-grained species recognition is a major focus in ecology and biodiversity monitoring, which poses unique challenges (e.g., high intra-class variance, inter-class similarity, and class imbalance) Nadarajan et al. (2009); Boulais et al. (2021); Gilby et al. (2017); Frances-cangeli & Jacopo (2023); Conservancy (2017); Van Horn et al. (2018); Zhuang et al. (2020). For

Table 1: Comparison with existing datasets for fish recognition tasks. FishNet++ provides textual descriptions for more than 35,000 species, while previous datasets only provide species labels. FishNet++ supports additional tasks for detection, key-part localization, and segmentation.

| Datasets | Properties | | | Tasks | | |
| --- | --- | --- | --- | --- | --- | --- |
| | Images | Species | Descriptions | Open-Vocabulary | Detection | Part-Location |
| Fish4-Knowledge-A Nadarajan et al. (2009) | 27,370 | 23 | 0 | ✗ | ✓ | ✗ |
| SEAMPD21 Boulais et al. (2021) | 28,328 | 130 | 0 | ✗ | ✗ | ✗ |
| Fish-gres Chastine (2021) | 3,248 | 8 | 0 | ✗ | ✗ | ✗ |
| Mediterranean Fish Species Georgiou (2021) | ≈40,000 | 20 | 0 | ✗ | ✗ | ✗ |
| Fish Abundance Gilby et al. (2017) | 4,909 | 50 | 0 | ✗ | ✗ | ✗ |
| Image Dataset Francescangeli & Jacopo (2023) | 33,805 | 30 | 0 | ✗ | ✗ | ✗ |
| NCFM Conservancy (2017) | 16,915 | 8 | 0 | ✗ | ✗ | ✗ |
| iNaturalist_Fish iNaturalist (2021) | 54,006 | 369 | 0 | ✗ | ✗ | ✗ |
| WildFish++ Zhuang et al. (2020) | 103,034 | 2,348 | 0 | ✗ | ✗ | ✗ |
| FishNet Khan et al. (2023) | 94,532 | 17,357 | 0 | ✗ | ✓ | ✗ |
| **Ours** | 99,556 | 17,393 | 35,133 | ✓ | ✓ | ✓ |

aquatic environments specifically, new datasets have been released. These include Fishnet Open Images Database Kay & Merrifield (2021), an open images dataset of $86,000$ of fish from $34$ species taken from vessel-bourne cameras, which highlights domain conditions like murky water, skewed species distribution, and occlusion. AutoFish Bengtson et al. (2025), another dataset with $1,500$ controlled-setup collected images of $454$ fish instances annotated with segmentation and IDs. We compare FishNet++ dataset with further existing datasets in Table 1.

**MultiModal Large Language Models (MLLMs).** MLLMs have advanced multimodal understanding and reasoning through large-scale pretraining, supervised fine-tuning, and often RLHF OpenAI (2024a;b); AI (2024); DeepSeek-AI et al. (2025); Qwen et al. (2025); Jiang et al. (2024); Touvron et al. (2023); Yang et al. (2024); Ouyang et al. (2022). Scaling models and data has been key to their success, yet they still struggle with long or complex contexts Yin et al. (2024). To address this, retrieval-augmented generation(RAG) Lewis et al. (2020); Izacard & Grave (2021) approaches have emerged as a practical solution, enabling models to access and reason over extended external information while reducing hallucinations and improving factual grounding. Recent works like Mallen et al. (2023); Asai et al. (2023) extend RAG to long-form reasoning, multi-hop QA, and vision-centric tasks, e.g., MuRAG Chen et al. (2022b) with image-text memory banks. In this work, we also show RAGs as a potential approach to enhance the performance of MLLMs for the open-vocabulary recognition task.

## 3 FISHNET++

While it is estimated that over 95% of the world's bird species have been described Barrowclough et al. (2016), the vast majority of marine life remains a mystery, with some estimates suggesting over 90% of species are yet to be discovered Mora et al. (2011). Despite this enormous knowledge gap, the focus of the computer vision community has predominantly been on terrestrial animals Wah et al. (2011); Berg et al. (2014); Van Horn et al. (2015). To help bridge this disparity and advance aquatic science, we introduce FishNet++, a large-scale, multi-modal benchmark developed from the original FishNet dataset Khan et al. (2023). Our primary goal is to enable the development of models capable of large-scale, language-based species recognition, a foundational step towards the ultimate challenge of identifying unseen or newly discovered species.

FishNet++ is enriched with 35,133 textual species descriptions and annotations for detection and key-part localization. We outline our comprehensive data collection methodology below, which includes a rigorous process for taxonomic correction, description generation, and the collection of bounding box and key-point annotations. To ensure the scientific validity of our benchmark, this entire process was conducted in close collaboration with experts in marine biology.

### 3.1 SPECIES DESCRIPTION

To generate descriptive text for each species, we first identified multiple reliable sources to serve as our knowledge base. FishBase Froese & Pauly (2025) was used as the primary source of morphological and ecological information, and supplemented by iNaturalist iNaturalist contributors (2025),

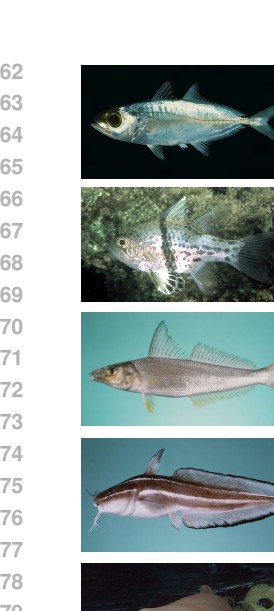

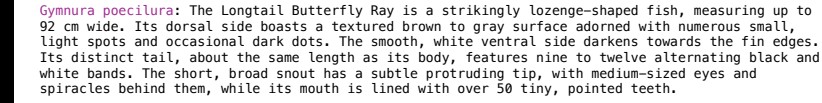

**Trachurus indicus**: The Arabian Scad, or Trachurus indicus, features an elongated, slightly compressed body that reaches up to 35 centimeters in length. Its upper body displays dark dusky hues, transitioning to nearly black or greenish-blue, while the flanks and belly gleam in silvery to white tones. A distinctive black spot adorns the upper margin of its operculum. Its moderately large eyes, covered by a well-developed adipose eyelid, sit above a broad jaw filled with small teeth. The fish boasts two dorsal fins, with the first showcasing eight spines, enhancing its streamlined shape.

**Sphaeramia orbicularis**: The orbiculate cardinalfish, or Sphaeramia orbicularis, is a small fish, reaching up to 10 cm with a short, deep, and compressed body. Its color is a greenish-grey, adorned with a prominent dark vertical 'waistband' across its body, and scattered dark spots decorating its head and fins. The pelvic fins are notably dark. It features 8 dorsal spines and 9 soft rays, alongside 2 anal spines and 9 soft rays. This fish typically inhabits shallow, reef-associated waters, creating small groups among mangroves and rocky debris.

**Sillago sihama**:The Silver Sillago is an elongated fish with a streamlined body and a circular cross-section, typically reaching up to 31 cm in length. Its smooth, ctenoid scales shimmer with a silvery hue, reflecting light beautifully. The head is straight, with normal-sized eyes and a terminal mouth. It features 11 to 13 spiny dorsal fins, complemented by soft rays, and a forked caudal fin that aids in agile swimming. Found in coastal waters, this fish often buries itself in sand, exhibiting a blend of grace and camouflage in its natural habitat.

**Plotosus lineatus**:The Striped Eel Catfish (Plotosus lineatus) is a slender, elongated fish reaching up to 32 cm in length. Its body showcases a rich brown hue adorned with striking cream or white longitudinal bands that run from head to tail. The fins are uniquely fused, giving it an eel-like appearance, while the first dorsal and pectoral fins boast sharp, venomous spines. Surrounding its wide mouth are four pairs of sensitive barbels, enhancing its tactile senses. This fish's combination of color, shape, and formidable spines makes it a distinctive inhabitant of coral reefs and estuaries.

**Carcharias taurus**: The Carcharias taurus, or sand tiger shark, exhibits a fusiform body shape with placoid scales. It is characterized by a short, pointed snout, small eyes, and protruding spike-like teeth. Its dorsal and anal fins are small and equal-sized, with the first dorsal fin closer to the pelvic than pectoral fins. The heterocercal caudal fin has a pronounced subterminal notch and a short ventral lobe. The shark is pale brown or grey, paler underneath, with dark spots that fade in adults, and plain fins.

**Gymnura poecilura**: The Longtail Butterfly Ray is a strikingly lozenge-shaped fish, measuring up to 92 cm wide. Its dorsal side boasts a textured brown to gray surface adorned with numerous small, light spots and occasional dark dots. The smooth, white ventral side darkens towards the fin edges. Its distinct tail, about the same length as its body, features nine to twelve alternating black and white bands. The short, broad snout has a subtle protruding tip, with medium-sized eyes and spiracles behind them, while its mouth is lined with over 50 tiny, pointed teeth.

Figure 1: Examples of species description summarized by GPT-4o OpenAI (2024a) using information scraped from credible sources as described in Section 3.1.

WoRMS WoRMS Editorial Board (2025), and NOAA NOAA Fisheries (2025). For species available on FishBase, $21,279$ out of the known $35,133$ fish species, we extracted detailed morphological data directly. For the remaining $13,854$ species missing morphological information from FishBase, we crawled iNaturalist, WoRMS, NOAA, and other supplemental sources to collect all available information. We then used GPT-4o OpenAI (2024a) to consume the information and produce a coherent and concise descriptive summary of each species. To validate the reliability of the generated descriptions, a subset of fifty descriptions was examined by experts, confirmed to be of reasonable accuracy and to be visually discriminative within the constraints of the description parameters (i.e., coherent and concise).

We also evaluated the description in a user study. The users are shown four images of the corresponding species along with the description, and they are asked to rate the description on a 1-5 scale, with 1 indicating "not helpful at all" and 5 indicating "very helpful" for identifying the species. This was done for 1,000 marine species descriptions. Each description was rated by three human annotators. The descriptions received a mean score of 3.9, a median of 4.0, and a mode of 4.0, highlighting that the descriptions are of good quality for recognition.

## 3.2 KEY-POINT SELECTION AND COLLECTION

We finalized six-part locations and one attribute to be collected for every image in FishNet++. The parts are as follows: 1) Eye location, 2) Mouth location, 3) Pectoral, pelvic, and anal fin location, 4) Center of the main body, 5) Tail (caudal fin) start, and 6) Tail end. All the parts were annotated by pixel location in each image. Additionally, we record whether the species is underwater or above water. Fin locations may involve multiple points depending on the number of fins, with variations by species, and are subject to image angle and occlusion. Similarly, the apex of the tail can have one or two location points depending on the shape of the tail. The selection of these parts and attributes was done in consultation with experts to ensure the dataset's utility for both the machine learning and aquatic science communities. A key piece of information provided by the key points is the aspect ratio, which has been linked to species' behaviour, metabolism, ecological

Figure 2: Example images from FishNet++ showcasing part-level annotations. Each keypoint is color-coded by semantic part: eye (orange), fins (blue), mouth (magenta), body center (yellow), tail start (green), and tail apex (red). The number and placement of fins vary across species, and some species exhibit a forked tail apex. For each image, we also display the annotated bounding box.

lifestyle, and response to thermal stress Sambilay (1990); Campos et al. (2018); THOMSON & SIMANEK (2015). This information can therefore be valuable in understanding species' ecology and can contribute to conservation decision-making. Additionally, key-part location can serve as weak supervision to obtain dense annotations like segmentation. Further discussed in Section D.

To collect the part location annotations, we partnered with a company specializing in data annotations. Experts supervised and validated the annotation process to ensure quality control. Once the annotators were familiar with the process, we implemented a system of regular manual checks to maintain the quality of the part location annotations. FishNet++ includes $86,589$ instances of eye locations, $77,990$ instances of mouth locations, $281,426$ instances of fin locations, $80,653$ instances of body locations, $73,785$ instances of tail-start locations, and $105,983$ instances of tail-end locations. In total, we provide $706,426$ key part locations for our dataset. From these images, $38,326$ images of fish are above the water surface.

## 3.3 TAXONOMIC CORRECTIONS

The taxonomy of species around the world is continuously evolving Bouchet et al. (2023), making it essential to ensure that datasets reflect the most up-to-date and accurate species names. During our analysis, we found that 266 species names from the FishNet Khan et al. (2023) dataset no longer aligned with current taxonomic standards (as per Froese & Pauly (2025)). To address this, we manually remapped these outdated names to their correct, updated counterparts. Following this, we also associated each species in our dataset with its corresponding species code from FishBase, which remains the same even as taxonomic names change. This provides a straightforward mechanism for keeping our dataset aligned with current taxonomic nomenclature. In the end, we identified 36 images that did not correspond to any known species from the entire taxonomy. These may represent entirely new species to science.

## 3.4 ADDITIONAL IMAGES

The original Fishnet Khan et al. (2023) is highly long-tailed, with only 495 species with ten or more images. The bias in the number of images is often associated with those species that are not exploited commercially at large scales (either for fisheries, ecotourism, or the aquarium trade), those that are found in less well-researched parts of the world, or those found in less accessible depth ranges. For FishNet++, we sought to increase the number of species with reasonable image representation. We collected additional images for species from various underrepresented regions worldwide, including Egypt, Indonesia, Oman, Seychelles, Papua New Guinea, and Saudi Arabia, sourced through a wider network of collaborators who provided access to their extensive collections. In total, we gathered an additional 5,024 images, increasing the number of species with at least ten images from 495 to 804, significantly enhancing its diversity and representation.

# 4 EXPERIMENTS

Based on FishNet++, we first evaluate the performance of various VLMs and MLLMs on the task of fish recognition. This is followed by a thorough analysis to explain the poor performance.

## 4.1 DATA SPLITS

We follow a 75-5-20 train-validation-test split strategy for species with a sufficient number of images. Specifically, for species with at least 5 images, 75% of the images are used for training, 5% for validation, and the remaining 20% for testing. For species with 3 or 4 images, we assign one image to the test set and use the remaining images for training. Species with fewer than 3 images (i.e., only 1 or 2 images) are not included in the main split. Instead, these rare cases are grouped into a separate "rare split", which exclusively contains species represented by 1 or 2 images. This splitting strategy is inspired by FishNet Khan et al. (2023), which drops species with very few samples (1 or 2) for the classification experiments. However, in contrast, we retain these underrepresented species in the rare split to thoroughly evaluate the recognition capabilities of vision-language models. The test set contains $15,518$ images, while the rare set contains $16,367$ images. The frequent set consists of $5,584$ species, and the rare set consists of $11,809$ species.

## 4.2 RECOGNITION RESULTS

Unlike traditional classification tasks that rely on a closed and predefined label space Wu et al. (2024), this task operates under an open and continually expanding set of species labels. To address this challenge, we leverage Vision-Language Models (VLMs) and MLLMs while utilizing all $35,133$ textual descriptions of species to infer the species present in the image. For CLIP-based VLMs Radford et al. (2021); Ilharco et al. (2021); Zhai et al. (2023), the approach is straightforward: we compute the cosine similarity between the visual embedding of an input image and the text embeddings of species descriptions. When species descriptions exceed the model's context length, we chunk them appropriately. The species whose

Table 2: Classification Accuracy: Evaluation of various open-source VLMs and MLLMs on the fish species open-vocabulary recognition task from species descriptions. Highest performance is in bold, and second-highest is in underline.

| Method | Frequent Species | Rare Species |
|---|---|---|
| OpenCLIP Ilharco et al. (2021) | 1.0 | 0.2 |
| BioCLIP Stevens et al. (2024) | 2.3 | 0.2 |
| CLIP Radford et al. (2021) | 2.4 | 0.2 |
| SigLIP Zhai et al. (2023) | 2.6 | 0.5 |
| LLaVa-Next Liu et al. (2023a) | 0.3 | 0.1 |
| LLaVaOne Li et al. (2024) | 0.6 | 0.0 |
| MiniCPM-V-2.6 Yao et al. (2024) | 1.7 | 0.1 |
| InternVL-2.5 Chen et al. (2024) | 2.0 | 0.0 |
| Pixtral-12b Agrawal et al. (2024) | 3.6 | 0.1 |
| Gemma-3 Team et al. (2025) | 5.5 | 0.2 |
| Qwen2.5-VL Qwen et al. (2025) | 6.2 | 0.2 |
| GPT-4o | **17.9** | **1.2** |

description yields the highest similarity is selected as the predicted label. To evaluate MLLMs, we formulate the task as a "Question Answering" task, where the question is to identify the species present in the image. We compare CLIP Radford et al. (2021), OpenClip Ilharco et al. (2021), Bio-Clip Stevens et al. (2024), and SigLip Zhai et al. (2023) as our VLM baselines. For MLLMs, we include InternVL-2.5(8B) Chen et al. (2024), MiniCPM(8.1B) Yao et al. (2024), Gemma-3(12.2B) Team et al. (2025), Pixtral-12b(12B) Agrawal et al. (2024), LlaVa-Next(13.4B) Liu et al. (2023a), LlaVaOne(8.03B) Touvron et al. (2023), and Qwen2.5-VL(8.29B) Qwen et al. (2025). We also include GPT-4o OpenAI (2024a) as a representative closed-source model.

As shown in Table 2, all models face significant challenges in accurately recognizing fish species from images, highlighting the difficulty of fine-grained open-world classification in the marine domain. Among all open-source models, Qwen2.5-VL achieves the highest performance on frequent species, followed by Gemma-3, while SigLIP performs best on the rare species subset. Although the overall accuracy remains low, it is still three orders of magnitude better than random guessing, highlighting the models' ability to learn some meaningful signal despite the task's difficulty.

### 4.2.1 RESULTS AT GENUS LEVEL

The species-level results indicate that current models do not yet achieve a strong overall performance, highlighting the difficulty of fine-grained, open-vocabulary classification. To investigate whether this challenge is alleviated at coarser taxonomic levels, we analyze whether the species predicted by the models belong to the correct genus.

In Table 3, we report the genus accuracy for all the models. We calculate the genus accuracy by mapping all species-level predictions made by each model to their corresponding genus. This allows us to distinguish between fine-grained misclassifications within the same genus and truly incorrect predictions across unrelated taxa. Compared to species-level results, we observe a clear improvement in accuracy, indicating that while models struggle with the extreme fine-grained species classification, they often predict the correct genus. We further extend this analysis to the family-level taxonomy in Section B. The performance improves substantially at the Family level, with Qwen2.5-VL and GPT-4o achieving 30.5% and 53.6% accuracy for frequent species, and 14.3% and 37.4% for rare species, respectively.

Table 3: Classification Results at the Genus Level. Highest performance is in bold, and second-highest is in underline.

| Method | Frequent Species | Rare Species |
|---|---|---|
| OpenCLIP Ilharco et al. (2021) | 5.0 | 2.3 |
| BioCLIP Stevens et al. (2024) | 8.5 | 3.2 |
| CLIP Radford et al. (2021) | 9.4 | 3.8 |
| SigLIP Zhai et al. (2023) | 14.8 | 8.6 |
| LLaVa-Next Liu et al. (2023a) | 5.6 | 0.7 |
| LLaVaOne Li et al. (2024) | 2.5 | 0.6 |
| MiniCPM-V-2.6 Yao et al. (2024) | 6.0 | 1.3 |
| InternVL-2.5 Chen et al. (2024) | 6.8 | 0.7 |
| Pixtral-12b Agrawal et al. (2024) | 8.2 | 3.0 |
| Gemma-3 Team et al. (2025) | 14.3 | 3.0 |
| Qwen2.5-VL Qwen et al. (2025) | 18.2 | 5.1 |
| GPT-4o | 34.2 | 14.1 |

## 4.3 DOMAIN KNOWLEDGE

To investigate whether the poor performance of MLLMs stems from a foundational knowledge gap. We devised a bidirectional name translation task using the common and scientific names for all 35,133 species, sourced from FishBase Froese & Pauly (2025). We evaluated the top-performing open source MLLM (Qwen2.5-VL) on its ability to map a scientific name to any of its corresponding common names, and conversely, a common name to its single correct scientific name. As shown in Table 4, the model struggles significantly with this task for marine species, with a mere 3.6% correct translations. In stark contrast, the same evaluation performed on the CUB-200-2011 bird dataset Wah et al. (2011) yielded an accuracy of 40.0%. This discrepancy strongly suggests that the model's failure is not a general limitation but lacks the basic taxonomic information needed to link common and scientific names, a task that requires no visual understanding.

Table 4: Performance of MLLMs on bidirectional name task.

| Method | Common → Scientific | Scientific → Common |
|---|---|---|
| Qwen2.5-VL | 3.6 | 3.6 |

## 4.4 VISUAL DOMAIN KNOWLEDGE

Having established the MLLM's semantic knowledge deficit with the name translation task, we next investigated if this was compounded by a failure in visual perception. For this, we designed a species verification task where the model was given an image and asked if a candidate species was present in the image or not. The task was repeated twice, once with the correct candidate and once with the wrong candidate. The wrong candidate was chosen either at random or was chosen from the nearest neighbors of the correct candidate in the CLIP space.

Table 5: Confusion matrix for the fine-grained differentiation task

| Method | Correct Species (Positive Case) | | Incorrect Species (Negative Case) | |
|---|---|---|---|---|
| | TP Rate (%) | FN Rate (%) | TN Rate (%) | FP Rate (%) |
| Qwen2.5-VL(random) | 81.4 | 18.6 | 67.1 | 32.9 |
| Qwen2.5-VL(fine-grain) | 56.4 | 43.6 | 34.8 | 65.2 |
| Random Chance | 50.0 | 50.0 | 50.0 | 50.0 |

From Table 4, it is clear that Qwen2.5-VL can distinguish if the species is present or not when the candidate is chosen at random, but when the candidate is more fine-grained, the model mostly answers 'Yes'. The average performance of Qwen2.5-VL for the fine-grained case is slightly worse than random chance. The poor performance on the fine-grained case reveals a failure in visual domain knowledge. Its inability to reliably accept the correct species and, crucially, reject the visually similar incorrect species, demonstrates that the issue is twofold. It not only lacks the deep domain knowledge to understand the subtle differences between species but also the fine-grained perceptual ability to discern those differences in an image. This shows the model's knowledge gap is not purely abstract but is also related to its visual processing capabilities. However, this test does not distinguish between coarse and fine-grained perception. To investigate this, we next evaluate the models on object detection and key-part localization.

Table 6: Performance of Qwen2.5-VL on coarse-grained detection. The performance is reported using IoU thresholds (50–90).

| Method | Frequent Species (%) | | | | | Rare Species (%) | | | | |
|---|---|---|---|---|---|---|---|---|---|---|
| | IoU50 | IoU60 | IoU70 | IoU80 | IoU90 | IoU50 | IoU60 | IoU70 | IoU80 | IoU90 |
| YOLO-12 Tian et al. (2025) | **95.2** | **92.0** | **84.7** | **67.9** | **35.2** | **95.6** | **93.3** | **88.1** | **74.1** | **40.6** |
| Qwen2.5-VL Qwen et al. (2025) | 91.5 | 85.1 | 73.8 | 54.8 | 26.7 | 95.2 | 91.0 | 82.0 | 63.8 | 31.3 |

Table 7: Performance of Qwen2.5-VL and a YOLO-based baseline on the fine-grained vision task.

| Method | Frequent Species (%) | | | | | | Rare Species (%) | | | | | |
|---|---|---|---|---|---|---|---|---|---|---|---|---|
| | Tail End | Fin | Tail Start | Body | Mouth | Eye | Tail End | Fin | Tail Start | Body | Mouth | Eye |
| YOLO-12 Tian et al. (2025) | **30.8** | **16.6** | **46.6** | **45.9** | **45.7** | **44.7** | **29.7** | 15.2 | **46.1** | **46.4** | **45.8** | **43.3** |
| Qwen2.5-VL Qwen et al. (2025) | 23.4 | 15.6 | 21.8 | 36.8 | 27.5 | 27.1 | 26.1 | **16.6** | 22.3 | 37.4 | 26.4 | 27.2 |

## 4.5 PERCEPTION CAPABILITIES

**Coarse-Grained Vision**: Before fine-grained recognition, a model must first perform coarse-grained visual localization, that is, correctly identifying the object's location within an image. Failure at this initial stage makes recognition unlikely. To assess this capability, we evaluated Qwen-VL on a detection task, using the ground-truth bounding box coordinates from FishNet++. The task is relatively straightforward, as our dataset predominantly contains single-instance images.

In Table 6, we compare Qwen2.5-VL with YOLO-12 Tian et al. (2025), trained on FishNet++. While Qwen2.5-VL underperforms YOLO-12, its results show a strong ability to localize fish, suggesting that recognition failures stem less from object detection and more from knowledge gaps or limitations in fine-grained visual perception, which we investigate next.

**Fine-Grained Vision**: To test the fine-grained visual capabilities of the Qwen model, we evaluate it on the task of key-part localization, where the model is required to identify the precise locations of body parts. In Table 7, we report the PCK Novotny et al. (2018); Yang & Ramanan (2013) score, which measures the proportion of keypoints that lie within a certain distance from the ground truth relative to the object size. The results in Table 7 compare the Qwen model against a YOLO model trained on the FishNet++ training set. Unlike the detection task, Qwen performs significantly worse on key-part localization. This poor performance highlights a critical limitation, as many fish species closer in the taxonomic tree often differ only in subtle, fine-grained details, such as eye coloration, the exact number of fin rays, or the shape of the tail. Capturing and reasoning over such nuanced visual cues is essential for reliable species-level recognition. This experiment shows us that while MLLMs like Qwen demonstrate competence on coarse-level recognition, they lack the fine-grained perceptual sensitivity needed for domains such as biodiversity research.

## 4.6 DOES FINETUNING HELP?

To enhance the zero-shot recognition performance of Qwen2.5-VL, we explore multiple approaches.

1) **retrieval-augmented generation**: The model is provided with textual context from species descriptions, but RAG performance depends heavily on the retrieval step. Since this is a cross-modal task, VLMs like CLIP are natural candidates, yet they perform poorly (see Table 2). To improve retrieval, we use an ensemble of CLIP, BioCLIP, and SigLIP, which yields more accurate candidates. The top 10 retrieved species descriptions are then passed to Qwen2.5-VL as context. This ensemble-based RAG (E-RAG) improves performance by $\approx 1\%$ (see Table 8), with potential for further gains using stronger retrievers. Retrieval results for individual VLMs are detailed in Section C.

2) **supervised-finetuning**: To enhance recognition performance, we performed LoRA-based supervised finetuning (SFT) on the Qwen-VL model using our proposed training set. This process substantially improved accuracy on frequent species, from a zero-shot baseline of 6.2% to 37.0%. We compare this against two strong baselines: pre-trained ViT Dosovitskiy et al. (2021) and BEiT Wang et al. (2023) finetuned on FishNet++. While our finetuned Qwen-VL outperforms the ViT baseline, BEiT achieves the highest accuracy at the species level. However, an analysis at higher taxonomic levels reveals that the finetuned Qwen-VL surpasses BEiT. This suggests that while BEiT may over-

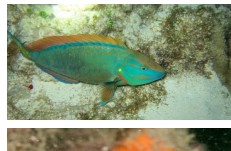
Sparisoma viride: The fish in the image aligns with *Sparisoma viride* due to its fusiform, oval body shape and convex forehead, consistent with the species' description. Its greenish-blue coloration matches the terminal phase male's hue, while the faint yellow gill spot and curved caudal fin mark suggest sexual dimorphism typical of advanced phases. Cycloid scales, though not distinctly visible here, are characteristic of the species. The protogynous nature (initially female) may explain the subdued pigmentation compared to vivid terminal phase males, but key morphological traits confirm its identification.

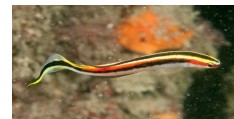
Trachinops taeniatus: The fish in the image aligns with *Trachinops taeniatus* due to its elongated, sleek body and prominent dorsal fin, both key traits described for this species. The vibrant coloration, featuring yellow, blue, and black hues with subtle patterning, matches the "striking blues, greens, and yellows" noted in its description. Additionally, the fish's streamlined shape and smooth texture reflect its adaptation to inshore reef environments, consistent with the species' habitat preferences. These visual characteristics collectively confirm its identification as the Eastern hulafish.

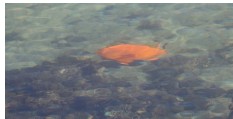
Hypsypops rubicundus: The fish in the image aligns with *Hypsypops rubicundus* (Garibaldi) based on key visual traits. Its vibrant orange coloration matches the species' typical hue, aiding immediate identification. The body appears short, deep, and compressed, consistent with the Garibaldi's stout shape, though the image's angle may obscure full depth. A convex forehead and terminal mouth are subtly suggested by the head's rounded profile and mouth placement near the snout. The caudal fin seems forked, aligning with the species' description, while dorsal and pectoral fins lack distinctive markings, as observed. These features collectively support the classification, despite limited visibility of scale texture or finer details.

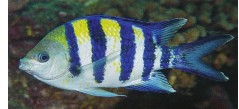
Abudefduf vaigiensis: The fish in the image matches *Abudefduf vaigiensis* (Indo-Pacific sergeant) based on key visual traits. First, its body is compressed, short, and deep, with a straight forehead and terminal mouth, consistent with the species' description. Second, the distinctive coloration includes five broad bluish-black vertical bars: the first and third are separated by a yellow patch near the head, aligning with courtship displays noted for this species. Third, the dorsal fin extends over most of the back, and the caudal fin is forked without dark bands, both characteristic of *A. vaigiensis*. These features collectively confirm the identification.

Figure 3: Qualitative examples of species identification and reasoning generated by our finetuned Qwen-VL model when trained for explainability.

fit to specific species-level features, Qwen-VL learns a more semantically robust representation, producing predictions that are taxonomically closer to the ground truth.

3) **Explainable supervised-finetuning**: To fully leverage FishNet++, we finetune Qwen to predict the correct species and generate supporting reasoning. This auxiliary task incurs minimal cost to recognition performance while greatly improving interpretability.

To construct the training corpus of reasoning, we employ GPT-4.0, which is provided with the input image(from the training set), the candidate species, and the species description, and asked to generate a concise justification for why the image corresponds to the given species. These reasoning texts are then paired with the species labels and used jointly during finetuning. We report the overall per-

Table 8: Finetuned classification results. Qwen2.5-VL ft. represents a finetuned version, and int. represents a finetuned version with reasoning.

| Method | Accuracy | | |
|---|---|---|---|
| | Species | Genus | Family |
| ViT | 25.3 | 31.5 | 38.4 |
| BeiT | **43.4** | 50.9 | 58.2 |
| Qwen2.5-VL | 6.2 | 18.2 | 30.5 |
| Qwen2.5-VL + RAG | 4.8 | 15.7 | 21.6 |
| Qwen 2.5VL + E-RAG | 7.1 | 22.7 | 46.2 |
| Qwen2.5-VL ft. | 37.0 | **51.5** | 64.7 |
| Qwen2.5-VL int. | 35.4 | 51.0 | **65.4** |

formance of Qwen under different training settings in Table 8, and additionally provide qualitative examples showcasing both predictions and their associated reasoning in Figure 3. Beyond accuracy, the generated explanations make the model's decisions more transparent and interpretable. Such interpretability is particularly valuable for marine scientists, as it enables verification of the model's decision-making process, facilitates error analysis when misclassifications occur, and provides human-readable insights that can support downstream ecological studies.

## 5 CONCLUSION

In this work,

1. We introduce FishNet++, a comprehensive multimodal benchmark for marine species recognition, designed to evaluate the strengths and limitations of MLLMs on fine-grained ecological tasks, offering textual descriptions, bounding boxes, and key-part annotations.

2. Our analysis reveals that state-of-the-art VLMs and MLLMs struggle with fine-grained taxonomic and morphological distinctions despite general recognition ability.

3. Through diagnostic experiments, we disentangle errors from domain knowledge gaps, weak visual perception, and limited reasoning.

4. Fine-tuning on FishNet++ narrows the performance gap, and explainable fine-tuning further boosts interpretability, underscoring the importance of domain-specific benchmarks.

## 6 ETHICS

The primary goal of this research is to advance AI for a positive societal impact, specifically in the domain of biodiversity conservation and marine biology. Our work introduces a new benchmark, FishNet++, which is constructed from publicly available images, a personal collection of images collected from collaborators throughout the world, and textual data sourced from encyclopedic resources like Wikipedia. All data used will be made publicly available, with an appropriate license. The dataset contains images of animal species and does not involve human subjects, thus presenting no personal data privacy concerns.

We acknowledge that all large-scale datasets are susceptible to inherent biases. Our benchmark may reflect geographic and taxonomic biases present in the publicly available data it is derived from. Similarly, the language models used for generating and distilling descriptions (e.g., GPT-4o) may carry their own latent biases. We have sought to mitigate this by involving aquatic science experts in our data curation process. We believe the potential for misuse of this technology is low, as its primary application is intended for scientific research and environmental monitoring.

## 7 REPRODUCIBILITY

We are committed to ensuring reproducibility of our work. All datasets, including the curated descriptions, will be made publicly available under appropriate licenses. For evaluation, we specify architectures and training details in the main text and supplementary material. Our codebase, including data loaders, evaluation scripts, and fine-tuning implementations for QWEN2.5-VL, will be released on GitHub. Random seeds are fixed in all experiments, and we report results across multiple runs where applicable. Together, these steps ensure that our results can be independently verified and extended by the community.

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

## A  EXPERIMENTAL DETAILS.

All inferences were performed on a single A100 GPU. For VLMs, the species prediction was made by selecting the class belonging to the chunk with the highest similarity score. For MLLMs, the species name was generated via prompting. We used two prompt variants: 1) without context: *Given the image of the fish, please answer with the species to which the fish belongs to? Only answer with the species scientific name.* and with RAG: *I have an image of a fish and need to identify its species type. I have narrowed it down to ten possible species. Please use the following descriptions to determine the most likely species: {}. Analyze the fish in the image, considering its physical characteristics, and compare them to the given species descriptions. Provide only the name of the most likely species.* In the RAG setting, we provided the MLLMs with the top 10 retrieved species descriptions. We also conduct an ablation study varying the number of descriptions fed to the MLLMs.

**Training Details** LORA-based supervised fine-tuning of Qwen was done on 4-A100 GPUs with 80GB memory for 4000 steps, with an effective batch size of 32, rank 8. Optimization was conducted using AdamW, employing an initial learning rate of 0.0001 with a cosine learning schedule and a 0.1 warmup ratio.

YOLO-based model training was performed using the Ultralytics YOLO framework for 30 epochs with a mini-batch size of 16 images. All input images were uniformly resized to 640×640 pixels. Optimization was conducted using Stochastic Gradient Descent on an NVIDIA V100 GPU with 32 GB of memory, employing an initial learning rate of 0.01, a momentum factor of 0.937, and a weight decay of 0.0005. A 3-epoch warmup phase was employed, linearly increasing the momentum from 0.8 and the bias learning rate from 0.1. We used the corresponding YOLO model as the base model with pre-trained weights utilized to speed up the convergence and enhance performance.

## B  HIGHER TAXONOMY RESULTS.

We extend our evaluation to the family-level classification, building upon the species and genus-level results presented in the main paper. From Table 9, we can see that as we go higher in the taxonomic hierarchy, from species to genus to family, the classification task becomes less granular, leading to improved performance across models. This trend is consistent with the inherent structure of biological taxonomy, where higher-level categories encompass broader groupings of organisms. Notably, the relative performance of models remains consistent across taxonomic levels. Qwen2.5-VL continues to outperform other open-source models, and its performance is further enhanced through the integration of the Ensemble RAG framework.

Table 9: Classification Accuracy: Evaluation of various open-source VLMs and MLLMs on the fish family open-vocabulary recognition task from species descriptions. Highest performance is in bold, and second-highest is in underline.

| | Frequent Species | Rare Species |
|---|---|---|
| OpenCLIP Ilharco et al. (2021) | 14.4 | 10.3 |
| BioCLIP Stevens et al. (2024) | 17.7 | 12.7 |
| CLIP Radford et al. (2021) | 22.7 | 15.8 |
| SigLIP Zhai et al. (2023) | 32.9 | 28.8 |
| LLaVa-Next Liu et al. (2023a) | 8.9 | 2.1 |
| LLaVaOne Li et al. (2024) | 6.7 | 3.4 |
| MiniCPM-V-2.6 Yao et al. (2024) | 13.1 | 5.4 |
| InternVL-2.5 Chen et al. (2024) | 12.9 | 2.9 |
| Pixtral-12b Agrawal et al. (2024) | 14.7 | 12.6 |
| Gemma-3 Team et al. (2025) | 24.6 | 12.5 |
| Qwen2.5-VL Qwen et al. (2025) | 30.5 | 14.3 |
| GPT-4o | **53.6** | **37.4** |

## C  VLM RETRIEVAL PERFORMANCE.

We report retrieval performance across both frequent and rare classes using Mean Reciprocal Rank (MRR) at 1, 5, and 10 in Table 10. Among individual models, SigLIP consistently performs the best, achieving an MRR@10 of 4.5 on seen classes and 1.2 on unseen classes. In contrast, BioCLIP, CLIP, and OpenCLIP show lower performance individually, with OpenCLIP performing the worst overall with MRR@10 of 1.6 and 0.4 on seen and unseen classes, respectively.

The best retrieval performance is observed when we combine all three models, CLIP + OpenCLIP + BioCLIP, achieving an MRR@10 of 8.4 on seen classes and 1.2 on unseen classes. This demonstrates that model ensembling can significantly boost retrieval quality, particularly for seen species.

Table 10: Mean Reciprocal Rank (MRR) at 1, 5, and 10 for retrieval performance on frequent and rare species. While individual models like SigLIP outperform others, the combination of CLIP, OpenCLIP, and BioCLIP yields the highest performance on seen classes. All models show a noticeable drop in performance on unseen species, clearly demonstrating the difficulty of generalization. Highest performance is in bold, and second-highest is in underline.

| | Frequent | | | Rare | | |
| --- | --- | --- | --- | --- | --- | --- |
| | MRR@1 | MRR@5 | MRR@10 | MRR@1 | MRR@5 | MRR@10 |
| BioCLIP Stevens et al. (2024) | 2.3 | 3.3 | 3.5 | 0.2 | 0.4 | 0.4 |
| CLIP Radford et al. (2021) | 2.4 | 3.5 | 3.8 | 0.2 | 0.4 | 0.4 |
| OpenCLIP Ilharco et al. (2021) | 1.0 | 1.5 | 1.6 | 0.2 | 0.3 | 0.4 |
| SigLIP Zhai et al. (2023) | 2.6 | 4.1 | 4.5 | 0.5 | **1.0** | **1.2** |
| **E-RAG** | **5.5** | **7.9** | **8.4** | **0.6** | **1.0** | **1.2** |

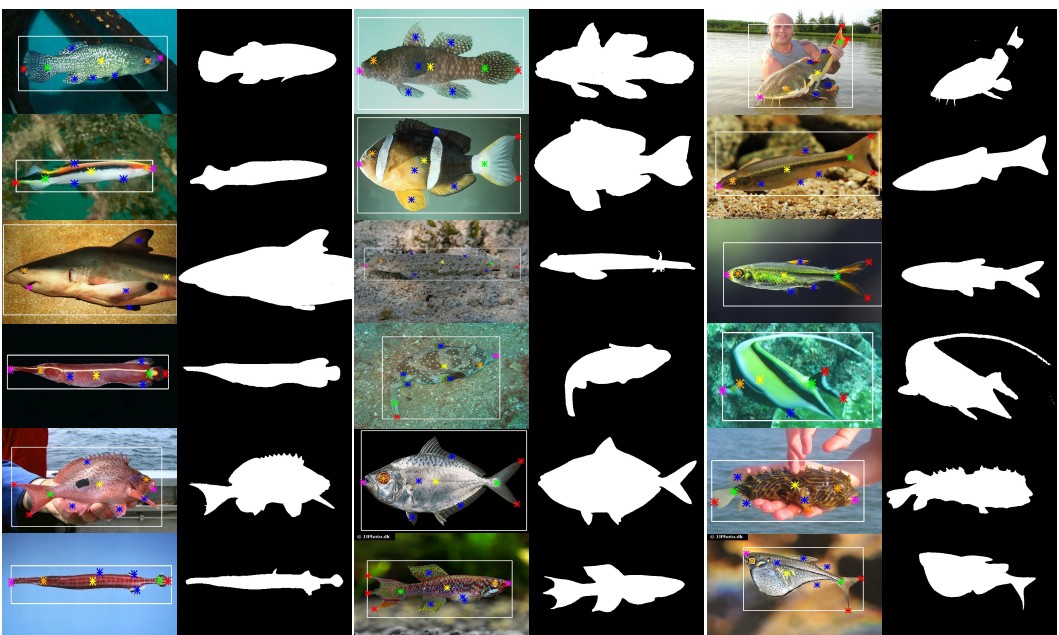

Figure 4: We show the same images from Figure 2 with segmentation masks obtained from our automated pipeline using key-points as supervision.

However, across all models, performance drops substantially on unseen classes, highlighting the challenge of generalization in open-world species retrieval.

# D  SEGMENTATION

Obtaining segmentation masks is often a time-consuming and labor-intensive task. To address this, we employed a semi-automated pipeline to generate segmentation masks for FishNet++. Specifically, we used keypoints collected for each image as prompts to guide the Segment Anything Model Kirillov et al. (2023a), enabling it to better infer the approximate structure of the target object. This keypoint-guided approach proved highly effective. Some examples are shown in Figure 4.

To evaluate the effectiveness of this approach, we performed a manual evaluation of the generated segmentation mask from both the frequent and rare species test sets. In this test, the annotators were asked if the given segmentation mask completely covered the marine species without missing any part of its body. From 31,885 images of the two test sets, 24,278(76%) were considered to be perfect by users, and the remaining ones captured most of the body but often missed parts like tails and fins, as shown in Figure 5. This clearly shows that our approach is highly effective for obtaining automated segmentation masks.

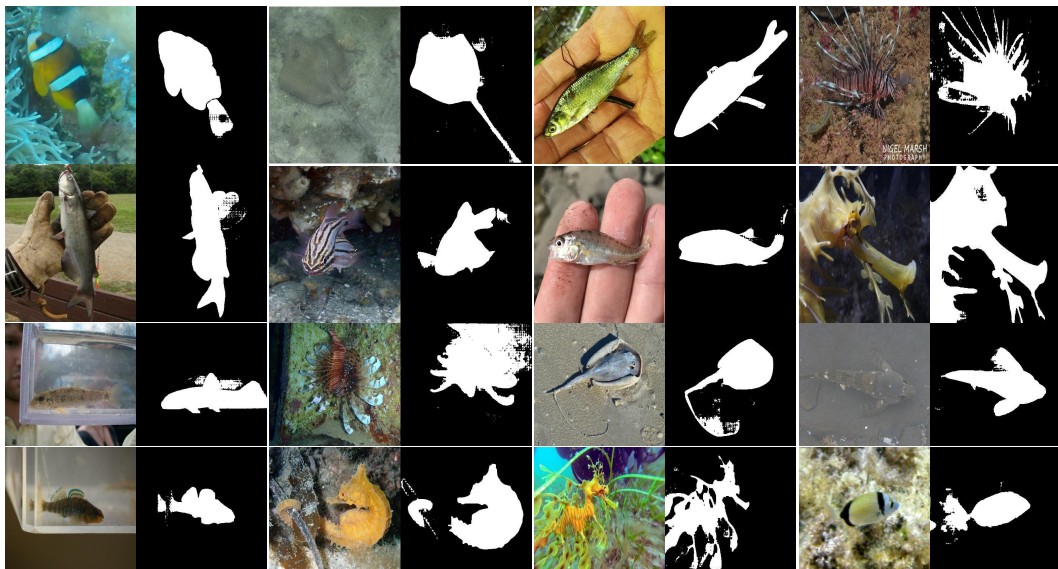

Figure 5: We show the samples with erroneous segmentation masks obtained from our automated pipeline using key-points as supervision.

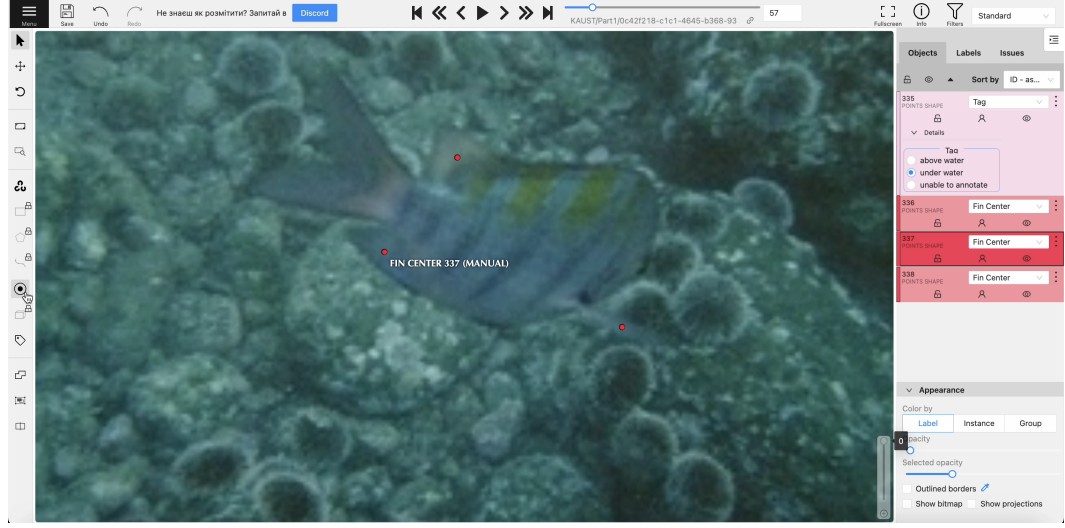

Figure 6: This figure illustrates the interface used for part location annotation. Here, the user has zoomed into the image to accurately label the center of the fin. Two fins have already been labeled, as indicated by the red-colored dot at the fin centers. The entire labeling process is efficient and user-friendly, as demonstrated in the video clip available here
.

## E    CROWDSOURCING DETAILS.

To enable efficient and accurate collection of data, we worked with an annotation service provider [1]. The custom-designed interface was developed to facilitate the collection and verification of part location and segmentation masks. We show the interface in Figure 6 and also include link to a video clip to completely demonstrate the annotation process.

---

[1]https://labelyourdata.com/

## F  LIMITATIONS.

Despite the extensive coverage and high-quality annotations provided by FishNet++, several limitations remain:

- While FishNet++ includes a large number of species and diverse annotations, the dataset is still constrained by available imagery. Certain ecological regions and rare species remain underrepresented, limiting the generalizability of models trained on this data to truly global scenarios that contain over 35,000 species.

- Prompt-based evaluation for MLLMs can be highly sensitive to the structure and content of the prompt, which may introduce bias in comparisons. Further, large models may hallucinate plausible but incorrect species names, particularly under open-vocabulary settings.

- Underwater imagery presents extreme domain shifts (lighting, turbidity, occlusion) that remain difficult for both MLLMs and task-specific models. Performance in these conditions, while informative, may not fully reflect real-time field performance.

