# OpenReview forum: "FishNet++: Analyzing the capabilities of Multimodal Large Language Models in marine biology."
_ICLR.cc/2026/Conference — Submitted to ICLR 2026_

### Official Review · Reviewer_pcRq · 2025-10-30

**Soundness:** 3
**Presentation:** 2
**Contribution:** 2
**Rating:** 4
**Confidence:** 4

**Summary:**

The authors introduce FishNet++, a comprehensive large-scale multimodal benchmark designed to evaluate the performance of MLLMs in domain-specific scientific contexts, particularly marine biology. The dataset contains over 99k images representing more than 17k fish species, accompanied by over 706k key-point annotations capturing morphological traits.

The study investigates the challenging problem of fine-grained fish species recognition, demonstrating that current state-of-the-art MLLMs significantly underperform in this specialized domain. To probe the sources of these limitations, the authors propose a diagnostic evaluation framework that decomposes model weaknesses into three distinct aspects, including domain knowledge, visual domain understanding, and perceptual capability. This framework includes tasks such as taxonomic name translation, species presence verification, and morphological key-point localization. Comprehensive analysis highlights the significant gap between general-purpose MLLMs and the expert-level understanding required for biodiversity research.

**Strengths:**

1) The paper is well-motivated, addressing a critical gap in the intersection between machine learning and marine biology. While large-scale multimodal datasets exist for terrestrial species, the marine domain remains underrepresented.

2) While prior datasets primarily focus on species-level classification, FishNet++ provides morphological annotations such as fin, eye, and tail key-points. This fine-grained information not only supports improved species recognition but also enables cross-disciplinary ecological analyses.

3) The authors systematically evaluate a wide range of VLMs and MLLMs, examining their recognition abilities across both frequent and rare species categories.

**Weaknesses:**

1) Despite its strong contributions, the paper’s diagnostic analysis is less comprehensive than claimed. While the authors claimed that a detailed diagnostic analysis of MLLMs' limitations as one of their contributions, the supporting evidence in Sections 4.3, 4.4, and 4.5 is not comprehensive enough to support this claim.

2) Authors primarily rely on a single model, Qwen2.5-VL for the whole analysis. While this model indeed performs best among the tested open-source models, using **only one representative MLLM to draw broader conclusions significantly weakens the generalizability of the claims**. Including a more diverse set of models, ideally at least four or five, would enhance the robustness of the diagnostic results and better substantiate that the observed failures are systemic rather than model-specific.

3) The evaluation of domain knowledge is also somewhat narrow. The bidirectional name translation task, though relevant, captures only a limited facet of scientific understanding. Domain knowledge in marine biology extends beyond nomenclature to encompass ecological and behavioral aspects such as geographic distribution, feeding patterns, and reproductive strategies, all of which are well-documented in public resources like FishBase. Incorporating such question-answering or retrieval-based assessments would provide a more holistic measure of domain-specific understanding and strengthen the conclusion that current MLLMs genuinely lack specialized knowledge.

4) The experiment designed to assess visual domain knowledge also feels reductive. It actually simplifies **the recognition task into a binary species verification task**, i.e., deciding whether a given image contains a candidate species. While informative, this setup does not fully capture fine-grained visual reasoning. To strengthen this component, the authors could explore other tasks, such as species re-identification, where it recognizes individuals of the same species under different poses, lighting, or angles based on consistent morphological alignments.

5) The perception experiments could benefit from more experimental details. Given that MLLMs often produce **unstructured or inconsistent outputs**, describing how the authors processed such results, handled format errors, and ensured objective evaluation would improve the transparency and reproducibility of the reported findings.

**Questions:**

Please refer to the weakness part.

---

### Official Review · Reviewer_yKN3 · 2025-10-31

**Soundness:** 2
**Presentation:** 2
**Contribution:** 2
**Rating:** 2
**Confidence:** 4

**Summary:**

This paper presents a refinement of an existing public, fine-grained image dataset of fish species. The authors added ~5000 images, semantic text descriptions, and keypoints. They then tested the performance of a suite of pretrained VLMs and MLLMs for species recognition, domain knowledge, and perception. They also fine-tuned a subset of those models for a recognition task. The pretrained models do not work very well and the fine-tuned ones do better. The most interesting result is that models like BioCLIP struggle with fish images.

**Strengths:**

- This paper tested a bunch of VLMs and MLLMs on a domain specific dataset. Their results on the domain knowledge probe are interesting, especially the visual test regarding species presence in an image using Qwen-2.5L.
- The addition of descriptions and keypoints to a marine biological dataset is a useful avenue of work.

**Weaknesses:**

The main source of novelty in this paper is the dataset. The experiments conducted are fairly standard assessments of model performance in a specialist domain. That does not have to be a weakness assuming the dataset is representative of something new and different.

I am not convinced FishNet++ clears that bar. The primary issue with the dataset is the relative size, it's focus on single instance images, and potential for bias. I think the authors need to dedicate more space to describing the image collection methodology and identifying sources of biases. They mention spatial biases only in a single bullet in the very last section of the supplement. I have many specific questions regarding the distribution of images and annotations, some outlined in the next section. Without such a treatment it is extremely difficult to assess how generalizable the results are to marine images or what the results of their tests might tell us about VLM and MLLM performance more broadly.

**Questions:**

- Where are the images from geographically? Are they collected from specific regions? What regions are left out? The authors list five locations for the new image data but do not specify what types of environments those places represent or how it compares to what was in the dataset already.
- How are the images collected in situ? From divers, towed cameras, autonomous vehicles?
- What depth range do the images represent?
- How was the dataset collated by the authors? Did they go into the field themselves? Webscraping? If the images were scraped, how much overlap is there between FishNet++ and other repositories?
- Given that the original FishNet is publicly available, what is the possibility it ended up in the training datasets of some of the pretrained model that were tested?
- What is the actual species distribution? The authors distinguish between rare and not rare using 3 images per class as the threshold. How long tailed is the distribution?
- How was the crowdworker-generated keypoint annotation data quality controlled? The authors describe contracting with a private company to get annotations from crowdworkers but do not indicate if any of that was validated by fish morphology experts.
- Who were the 'users' for the description evaluation articulated between 199 and 203? Are these people with training in marine biology or lay users?
- What are some of the limitations of the splitting strategy articulated in section 4.1? Why use these heuristics rather than splitting randomly or according to some relevant metadata?
- BioCLIP is the pretrained model tested that may directly include some of the taxonomic information required for these fish tasks. It'd be useful to expand more on that particular set of tests. Did the space of BioCLIP classes include some of the labels in FishNet? If so, that may warrant further evaluation.
- Can you speculate as to why GPT-4o did so much better than the other models tested in all the classification tests?

---

### Official Review · Reviewer_Em1W · 2025-11-01

**Soundness:** 2
**Presentation:** 3
**Contribution:** 2
**Rating:** 4
**Confidence:** 3

**Summary:**

The paper introduces FishNet++, a large-scale multimodal benchmark for marine fish recognition that augments prior resources with 35,133 textual species descriptions, 706,426 key-point annotations, and 119,399 bounding boxes across 99,556 images / 17,393 species. Using this benchmark, the authors show that state-of-the-art MLLMs and VLMs perform very poorly at open-vocabulary, fine-grained species identification (e.g., Qwen2.5-VL 6.2% on frequent species; GPT-4o 17.9%), while doing much better at coarser genus/family levels. They disentangle failure sources via tasks targeting (i) domain knowledge (common↔scientific name mapping), (ii) visual domain knowledge (species verification), and (iii) perception (detection and key-part localization), finding deficits in both taxonomic knowledge and fine-grained visual perception. The paper further shows that LoRA SFT on FishNet++ boosts Qwen2.5-VL from 6.2% → 37.0% species accuracy, with additional interpretability by training it to produce textual justifications; BEiT remains higher at the species level (43.4%), but Qwen improves more at higher taxonomic levels. The dataset construction describes taxonomic remapping, expert-validated descriptions, keypoint schema, and ethics/reproducibility commitments. Overall, the work positions FishNet++ as a diagnostic and advancement tool for aquatic biodiversity ML.

**Strengths:**

- FishNet++ fills a clear gap for marine fine-grained, open-vocabulary recognition with rich multimodal supervision (texts, boxes, keypoints) and careful taxonomic curation.
- Clear decomposition of failure modes (knowledge vs. perception) via targeted tasks (name translation, species verification, detection/keypoints) gives actionable insight, not just headline accuracies.
- Fine-tuning and explainable training meaningfully improve accuracy and produce human-readable justifications valuable to scientists.

**Weaknesses:**

- Species descriptions are generated with GPT-4o, and GPT-4o is also evaluated as a model—this could advantage GPT-4o (style/knowledge overlap) relative to open-source baselines; the paper does not quantify or control for this (e.g., re-running with human-only or held-out descriptions).
- Detection and keypoint experiments mostly compare a single MLLM (Qwen2.5-VL) to a YOLO baseline; including other MLLMs and modern detectors/pose estimators would strengthen conclusions about perception limits.
- For open-vocabulary classification, chunking of long descriptions, the exact QA prompts for each MLLM, and retrieval settings for (E-)RAG are only sketched, limiting reproducibility and making it hard to attribute performance gaps to retrieval vs. reasoning.
- Keypoint collection is expert-supervised, but there is no reported inter-annotator agreement, QC statistics by part, or error analysis by habitat/visibility (e.g., turbidity/occlusion), which are pivotal for fine-grained morphology tasks.

**Questions:**

- Can the authors report agreement metrics (IAA) and per-part error distributions for keypoints, and perhaps release a gold subset that is double-annotated for benchmarking?
- To address the GPT-4o confound, could they provide results with human-curated descriptions (or alternative LLM-authored descriptions held out from any evaluated model) and quantify the sensitivity?

---

### Official Review · Reviewer_ZqSX · 2025-11-01

**Soundness:** 2
**Presentation:** 3
**Contribution:** 2
**Rating:** 2
**Confidence:** 5

**Summary:**

The authors present FishNet++, a benchmark dataset that includes images of fish species, corresponding species names, textual descriptions generated by GPT-4o based on FishBase and other established biological knowledge bases, bounding boxes over entire fish, and manually annotated keypoints for multiple fish parts obtained via crowdworkers under expert supervision. Using this dataset, the authors evaluate the performance of multimodal large language models (MLLMs) on three tasks: fish species classification, fish bounding box detection, and keypoint-based localization of fish parts. The study provides insights into the current limitations of multimodal LLMs in fine-grained fish-part perception and demonstrates that fine-tuning with fish species descriptions can partially mitigate these issues.

**Strengths:**

- The paper presents a clear motivation for advancing research in marine species understanding and highlights challenges unique to fish images.

- The inclusion of manually annotated keypoints for multiple regions represents a commendable effort and could support detailed part-based analyses.

- Section 4.3 offers an interesting observation about multimodal LLMs’ difficulty in linking scientific names of fish species and their corresponding common names – an underexplored but relevant limitation for real-world biological applications

**Weaknesses:**

- Missing comparison with similar datasets. Introduction and Table 1 do not include comparisons with other recent datasets like Fathomnet **[A]**, Fish-Vista **[B]**, DeepFish **[C]**, etc..**[B]** also contains part-localization segmentation annotations of various fine-grained fish parts.

- Limited validation of GPT-4o generated species summaries. GPT-4o is used to summarize species information collected from FishBase and other knowledge bases, but there is no systematic evaluation of factual accuracy or hallucination. Manual validation was performed on a small subset (50 expert-checked and 1,000 user-study species), which seems insufficient to assess reliability at scale. Also, were the human annotators domain experts to understand biological terms like “placoid scales” (Figure 1- Carcharias taurus), “dorsal or pelvic fins”, etc?

- Questionable added value of text summarization. Looking at the FishBase dataset for the first example (Trachurus indicus) in Figure 1 (https://www.fishbase.se/summary/Trachurus_indicus.html), it shows that FishBase already contains textual information structured into various categories (e.g., environment, distribution, morphology, habitat). This already encodes rich and interpretable information. It is unclear whether converting them into free-form textual summaries using GPT-4o enhances usability for LLMs or simply compresses information that is better preserved in its original structured form.

- Lines 251–252 mention manual mapping of species names, but the process is not described. It would be helpful to know whether this mapping relied on an established taxonomy or biological knowledge base (e.g., Open Tree of Life or similar).

- Overlap with prior work that performs fish classification and fish part perception using multimodal LLMs: Fish species classification as a question-answering task using multimodal LLMs has been explored in VLM4Bio **[D]** and FishDetectLLM **[E]**. The perception capabilities of multimodal LLMs for fine-grained fish parts have also been explored in **[D]**, and additionally in **[B]**, with similar findings.

- The perception capabilities (Section 4.5) of MLLMs are only evaluated on a general-purpose multimodal LLM - Qwen, and are missing comparison to multimodal LLMs that are specifically trained for pointing (for example, Molmo **[F]**)

- The authors present performance improvement with fine-tuning LLMs and ViTs using the species descriptions that they curated. Similar fine-tuning has been done in **[E]** on the original FishNet dataset. This calls for a comparison of a) the authors’ dataset on the trained FishDetectLLM and b) how the authors’ findings from fine-tuning their models vary from the ones presented in **[E]**.

- In Figure 4 (Appendix D), it is unclear if using keypoints improves SAM masks vs only using bounding boxes to obtain SAM masks – i.e., what are the segmentation masks that were generated using bounding boxes only?

**References:**

**[A]** Kakani Katija, Eric Orenstein, Brian Schlining, Lonny Lundsten, Kevin Barnard, Giovanna Sainz, Oceane Boulais, Megan Cromwell, Erin Butler, Benjamin Woodward, et al. Fathomnet: A global image database for enabling artificial intelligence in the ocean. Scientific reports, 12(1):15914, 2022.

**[B]** Mehrab, Kazi Sajeed, et al. "Fish-vista: A multi-purpose dataset for understanding & identification of traits from images." Proceedings of the Computer Vision and Pattern Recognition Conference. 2025.

**[C]** Saleh, Alzayat, et al. "A realistic fish-habitat dataset to evaluate algorithms for underwater visual analysis." Scientific reports 10.1 (2020): 14671.

**[D]** Maruf, M., et al. "Vlm4bio: A benchmark dataset to evaluate pretrained vision-language models for trait discovery from biological images." Advances in Neural Information Processing Systems 37 (2024): 131035-131071.

**[E]** Zhu, Jiaxin, et al. ‘FishDetectLLM: Multimodal Instruction Tuning with Large Language Models for Fish Detection’. Knowledge-Based Systems, vol. 318, 2025, p. 113418, https://doi.org/10.1016/j.knosys.2025.113418.

**[F]** Deitke, Matt, et al. "Molmo and pixmo: Open weights and open data for state-of-the-art vision-language models." Proceedings of the Computer Vision and Pattern Recognition Conference. 2025.

**Questions:**

- There is a discrepancy between the number of species for which descriptions exist (more than 35,000 species) and the total number of species reported in Table 1 (17,393). It is unclear why descriptions exist for more species than are reported in the dataset.

- Lines 303-304: It is unclear how the species descriptions are chunked to obtain a single text embedding for CLIP. Also, how is evaluation done with BioCLIP? It is trained to classify using images and taxonomic levels.

- The authors present over 119k bounding boxes in the dataset (Lines 64-65), but it is unclear from the text how these bounding boxes were obtained.

- The paper states that keypoints were annotated for fins (pectoral, pelvic, and anal), but these are grouped under a single label category. Providing distinct class identifiers for each would enhance biological interpretability.

**Minor Comments:**

- In Section 4.4 Line 368, Table 4 is wrongly referenced instead of Table 5

**Details Of Ethics Concerns:**

No ethical concerns

---

### Meta-Review · Area_Chair_cv97 · 2026-01-07

**Summary:**

The paper introduces a large-scale multimodal benchmark dataset (FishNet++) for marine visual recognition and understanding. Features of this dataset include: fish images, corresponding species names, textual descriptions generated by GPT-4o, bounding boxes, and annotated keypoints for morphological trait analysis obtained via crowdsourcing.

While the reviewers appreciated the potential domain-specific contributions of the paper, their overall comments were largely on the negative side. Concerns were raised around: novelty/uniqueness of the dataset relative to existing similar datasets (such as DeepFish and FishBase); veracity of GPT-4o for generating text annotations; multimodal model comparisons (only Qwen-2.5-VL was considered, not others); and lack of additional details on the annotation crowdsourcing experiment. I agree that all these concerns are valid and need to be resolved prior to publication.

As an aside: despite the weaknesses I thought the paper was well-written and addresses an interesting problem. The authors are encouraged to take the reviewer feedback into account while preparing future revisions.

**Reviewer Concerns:**

NA (no rebuttal)

**Reviewer Scores:**

NA (no rebuttal)

---

### Decision · Program_Chairs · 2026-01-26

Reject